# Monitoring Pilot’s Mental Workload Using ERPs and Spectral Power with a Six-Dry-Electrode EEG System in Real Flight Conditions

**DOI:** 10.3390/s19061324

**Published:** 2019-03-16

**Authors:** Frédéric Dehais, Alban Duprès, Sarah Blum, Nicolas Drougard, Sébastien Scannella, Raphaëlle N. Roy, Fabien Lotte

**Affiliations:** 1ISAE-SUPAERO, Université de Toulouse, 31055 Toulouse, France; alban.dupres@orange.fr (A.D.); nicolas.drougard@isae.fr (N.D.); sebastien.scannella@isae.fr (S.S.); raphaelle.roy@isae.fr (R.N.R.); 2Department of Psychology, University of Oldenburg, 26122 Oldenburg, Germany; Sarah.Blum@Informatik.Uni-Oldenburg.de; 3Inria Bordeaux Sud Ouest, LaBRI, University of Bordeaux, Potioc Team, 33400 Talence, France; fabien.lotte@inria.fr

**Keywords:** dry-electrode EEG, real flight conditions, Artifact Subspace Reconstruction (ASR), oddball, auditory attention, neuroergonomics, mobi

## Abstract

Recent technological progress has allowed the development of low-cost and highly portable brain sensors such as pre-amplified dry-electrodes to measure cognitive activity out of the laboratory. This technology opens promising perspectives to monitor the “brain at work” in complex real-life situations such as while operating aircraft. However, there is a need to benchmark these sensors in real operational conditions. We therefore designed a scenario in which twenty-two pilots equipped with a six-dry-electrode EEG system had to perform one low load and one high load traffic pattern along with a passive auditory oddball. In the low load condition, the participants were monitoring the flight handled by a flight instructor, whereas they were flying the aircraft in the high load condition. At the group level, statistical analyses disclosed higher P300 amplitude for the auditory target (Pz, P4 and Oz electrodes) along with higher alpha band power (Pz electrode), and higher theta band power (Oz electrode) in the low load condition as compared to the high load one. Single trial classification accuracy using both event-related potentials and event-related frequency features at the same time did not exceed chance level to discriminate the two load conditions. However, when considering only the frequency features computed over the continuous signal, classification accuracy reached around 70% on average. This study demonstrates the potential of dry-EEG to monitor cognition in a highly ecological and noisy environment, but also reveals that hardware improvement is still needed before it can be used for everyday flight operations.

## 1. Introduction

Operating aircraft is a challenging activity that takes place under a dynamic and uncertain environment [1]. Pilots are required to perform multiple tasks such as controlling the aircraft trajectory, monitoring the flight parameters, performing check-lists, communicating with air traffic controllers, identifying potential threats (collision, failures and bad weather conditions) and potentially adapting the flight plan. The management of these multiple tasks can drastically increase workload [2,3,4] that in return may have a deleterious effect on their performance. Indeed, mental overload can alter visual [5] and auditory [6,7] attention, working memory abilities [8], the execution of actions [9], and lead to poor decision making [10]. One solution to improve flight safety and mitigate the occurence of human errors is to implement a passive brain computer interface (pBCI) [11,12,13] to continuously monitor mental workload [14] and to dynamically adapt pilot-cockpit interaction [15].

Electro-encephalography (EEG) represents one of the most popular technique in the pBCI community to infer mental workload. Generally, authors compute on-line frequency-domain features over the electrophysiological signal in the theta, alpha and beta frequency bands as neural markers of mental workload (for a review see [3]). Notably, transitions from low to high mental demand are associated with a decrease in the alpha band power over the parietal sites and an increase in theta band power over the frontal sites [16,17,18]. Other studies also reported increased beta [19] and gamma bands [20] power associated with higher task demands. Alternatively, time-domain analyses over the EEG signal can also underpin variations of mental workload using passive probe paradigms [21,22]. While participants are performing a primary task, their brain response to a secondary oddball task is assessed. In this latter task, participants are presented with rare target stimuli and frequent standard distractors that elicit event related potentials (ERPs) such as P300 (positive deflection around 300 ms) with higher absolute amplitudes for the deviant auditory stimuli than the standard tones [23]. The online estimation of ERPs is then used as an indirect index of mental workload amplitude as their amplitude decreases when the primary task demand increases [21,22]. However, most of the EEG-based pBCI to infer mental workload are conducted under laboratory settings and there is a need to test these neuroadaptive technologies in the real-world [24,25,26].

Since the pioneering work of Wilson et al. [27], EEG has been tested in actual flight conditions [28,29,30]. However, these authors used wet electrode EEG systems that might not be suitable for everyday flight operations as they require the use of conductive gel on the user’s scalp. Recent technological progress has allowed the development of gel-free pre-amplified dry-electrodes. Moreover, the use of a wireless communication protocol (e.g., Bluetooth, Wi-Fi) allows streaming electrophysiological data online, providing freedom of movement for the users and even enables signal processing during mobile recordings [31]. These mobile technologies open promising perspectives to measure the “brain at work” such as while operating aircraft. Nonetheless, the use of dry-electrode EEG remains challenging as it has a lower signal to noise ratio than classical wet/gel electrodes [32,33]. This issue might be particularly critical as the cockpit environment is known to be particularly noisy due to electromagnetic interferences (e.g., GPS antenna, radio communication), vibrations (e.g., engines), and pilots’ muscular activity.

Despite these technical challenges, some authors have tested dry EEG systems in actual flight conditions [34,35] and successfully implemented off-line pBCIs [36,37,38]. These studies used multiple channel (32 or 64) systems that are cumbersome and cannot be worn by subjects over long periods of time especially with the pilot’s headset on top of them. A relevant approach would consist in drastically reducing the number of electrodes for better comfort and potential integration in the pilot’s headset. The main drawback of reducing the number of electrodes is that it prevents from the use of efficient signal processing techniques such as independent component analysis (ICA) to identify artifactual components [39]. One solution [34] is to use Artifact Subspace Reconstruction (ASR), to automatically remove short-time high-amplitudes artifacts in EEG data [40]. ASR is a statistical method that learns properties of resting EEG during a calibration phase. This technique computes a principal component analysis on covariance matrices of the channel data to detect artifacts based on their statistical properties in the component subspace. Physiological signals like oscillations or event-related potentials will not deviate from a resting EEG signal by an amount that is detected by ASR as artifactual. Once an artifactual segment has been detected, it is reconstructed with estimated clean EEG data. Several studies have shown that ASR can be used to correct data during motion while retaining oscillations as well as other cortical signals of interest [41,42]. Recently, Blum et al. [43] developed and evaluated an adaptation of the existing ASR algorithm in which they explored whether the use of Riemannian geometry helps to attenuate typical EEG artifacts more reliably and with lower computational costs. In a comparison between ASR and Riemannian ASR (rASR), they found that rASR is faster and more sensitive to eye artifacts in mobile EEG data. At the same time, rASR preserves the signal of interest and improves its signal-to-noise ratio. rASR is an open source project and available as a Matlab toolbox [44].

In the present study, we test the feasibility to estimate pilots’ workload in real flight conditions with a highly portable six dry-electrode EEG system. The experimental scenario consists in performing two traffic patterns in two levels of flying difficulty (low and high load conditions). In the low load condition, the participant is monitoring the flight controlled by the flight instructor and in the high load condition, the participant is actually flying the aircraft. Along with the two flying tasks, pilots are presented with a passive auditory oddball. Our first objective is to determine whether we can extract frequency and time domain features over the EEG signals to discriminate the two flying conditions at the statistical level. Our second objective is to implement an off-line pBCI combining time and frequency-domain features to infer the participants’ mental workload [22,45].

## 2. Materials and Methods

### 2.1. Participants

Twenty-two visual flight rules (VFR) pilots (three women; mean group age: 25.4; mean flight hours: 40) completed the experiment. Pilots had normal or corrected-to-normal vision, normal hearing as attested by their medical clearance to fly. Pilots were instructed not to take any caffeine or medication prior to 24 h of the testing. The data from four participants were rejected due to data synchronization issues. Typical total duration of a subject’s session was about one hour. The experiment was approved by the European Aviation Safety Agency (EASA60049235). The methods were carried out in accordance with approved guidelines and participants gave their informed written consent.

### 2.2. Airplane

The study was conducted using the ISAE-SUPAERO (Institut Supérieur de l’Aéronautique et de l’Espace—French Aeronautical University in Toulouse, France) experimental light aircraft (see Figure 1). The DR400 light aircraft was powered by a 180 HP Lycoming engine and was equipped with classical gauges, radio and radio navigation equipment, and actuators such as rudder, stick, thrust and switches to control the flight.

### 2.3. Flight Scenario

The scenario consisted of two consecutive traffic patterns at Lasbordes airfield. Each traffic pattern, according to the standards of visual flight rules (VFR), is divided into five flight phases—the upwind take-off leg, the crosswind leg, the downwind leg, the base leg and the final landing (see Figure 1). In the first traffic pattern, defined as the low load condition, the participant (left-seated) was monitoring the flight controlled by the flight instructor (right-seated). In the second traffic pattern, defined as the high load condition, the participant was actually flying the aircraft and was supervised by the flight instructor. Each traffic pattern lasted around 500 s and the total experiment duration lasted around 20 min from take-off to parking. Along with the flying tasks (i.e., monitoring and flying), the participants were asked to perform an oddball paradigm with a total of 320 auditory stimuli: 25% were targets (80 normalized pure tone at 1100 Hz, 90 dB SPL) and 75% were non-targets (240 normalized pure tone at 1000 Hz, 90 dB SPL). Inter-trial interval was set to 2000 ms with a 1000-ms jitter. The audio stimuli were presented to the pilot’s aviation headset (Clarity AloftPro) through the auxiliary input of the experimental computer. The sound intensity of the stimuli and background environmental noise in the cockpit were measured with a sound level meter. As for a previous experiment [34], the sounds were presented at 85 dBA. The sounds of the stimuli were attenuated during radio communication but remained perfectly audible. The reported real-ear tested attenuation characteristics of the Clarity Aloft headset we used is reported to be 29 dB. The approximate signal to noise ratio was 35 dB with engine on. The experimenter was the backseater and his role was to place the EEG cap and trigger the oddball task. We used Lab Streaming Layer libraries (LSL, Swartz Center for Computational Neuroscience, UCSD, November 2018) to synchronize the oddball task in Matlab (Ver. 2017.b) with the Enobio acquisition software (NIC V2.0). Preliminary experiments were conducted with four pilots to pre-test the experimental scenario. NASA-TLX score confirmed that the two conditions elicited two different levels of mental workload (high load condition = 6.7, SD 0.45; low load condition 2.56, SD = 0.75).

### 2.4. EEG Analyses

#### 2.4.1. EEG Recording

EEG data were recorded at 500 Hz using the six dry-electrode Enobio Neuroelectrics system (Fz, Cz, Pz, Oz, P3 & P4 sites) positioned according to the 10–20 system. CMS and DRL were used as reference electrodes. The offset level for each subject was carefully checked and were within the margins recommended by Neuroelectrics before starting the experiment. Thirty seconds of cleaned signal was recorded on the ground before starting the experimental protocol while the participant was seated in the aircraft. These cleaned data were used for rASR calibration purpose. All the EEG analyses were ran using EEGLab (V14.1.2) and Matlab (17.b).

#### 2.4.2. EEG Pre-Processing

For the time-domain analyses, the continuous EEG data was filtered between 0.5–30 Hz (windowed-sinc FIR filter with an order of 250). Noisy portions of data (e.g., trials) were cleaned using the Riemannian ASR (rASR) version of the clean rawdata Matlab toolbox (see Figure 2). The toolbox contains the core functionality clean_asr to correct data segments which can be applied if short parts of the data is artifactual or only a minor portion of all channels is affected. Otherwise, additional functions of the toolbox remove noisy segments without providing a reconstruction of the data. The whole toolbox is available at [44], containing the Riemannian-adapted core functions together with all unchanged wrapper functions (parameters for clean_asr: flatline criterion = 5, highpass = [0.25 0.75], channel Criterion = 0.85, line Noise criterion = 4, burst criterion = 70, window criterion = 0.10).

The epochs for deviant and standard stimuli were extracted from the continuous data 0.2 s before and 1 s after stimuli onsets. The trials used for the ERP analyses were baseline normalized using data from 200 to 0 ms prior to the stimulus onset. A mean number of 17 trials per condition were dropped for each participant after processing the signal with rASR.

For the frequency-domain analyses, the continuous EEG data was high-passed (0.5 Hz) filtered and then noisy portions of data were removed using rASR plugin with the clean rawdata function (using the same parameters as described above). Frequency features were extracted using the Matlab function “cwt”, which computes a Morlet wavelet transformation to extract the spectral power of the delta [1–4], theta [4–8], alpha [8–12] and beta [12–16] bands (in Hz).

#### 2.4.3. EEG Statistical Analyses

Statistical analyses of the ERPs were carried out using a 3-way repeated measure analysis of variance (ANOVA) with load (Low, High), Type of sound (Frequent, Target) and Electrodes (Fz, Cz, Pz, P3, P4 and Oz) as within-subject factors. A two-way repeated measure ANOVA was carried out on the spectral band powers with frequency bands (delta, theta, alpha and beta) and electrodes (Fz, Cz, Pz, P3, P4 and Oz) as within-subject factors. We then apply the built-in EEGlab bootstrap test (10,000 iterations) for subsequent statistical analyses. Bootstrap resampling has advantages over parametric statistical tests in that it does not assume normal distribution and homoscedasticity of the value of interest or the error terms [46].

#### 2.4.4. EEG Processing for Single Trial Classification

Two different EEG processing pipelines were used for single trial classification: the first pipeline was synchronized on the stimuli onset in order to combine ERP and frequency based features (Figure 3), whereas the second pipeline computed only frequency features from continuous EEG (sliding window scheme), regardless of stimuli onset.

For both pipelines EEG processing was applied independently in order to simulate an online classification.

Concerning the first synchronized pipeline we extracted as many trials as stimuli. For each trial 2 s of epoch from 1 s before to 1 s after stimulus onset was extracted. Concerning the second pipeline, epochs were extracted from successive and non overlapping epochs of 2 s, regardless of stimuli onset. Then, for the two pipelines we applied an rASR filter on EEG signals from each epoch. The rASR filter was calibrated using the first 30 s of EEG recording, in order to use signals for calibration that were not used for the classification. After the rASR filtering step, the processing was different to compute ERP and frequency based features. Regarding ERP based features, EEG signals were band-pass filtered ([1 15 Hz]), using the EEGlab FIR filter (windowed sync FIR filter with hamming window and default parameters). Then, we removed the baseline from −0.2 s to 0 s according to stimulus onset, and again we extracted epochs from 0 to +0.6 s. Finally the signals were downsampled to 50 Hz. We then concatenated the resulting pre-processed EEG samples of amplitude values from all channels into a single feature vector for each epoch, and used this feature vector as input to the subsequent machine learning classifier (see below), as classically done for ERP classification in BCI [47]. Concerning frequency based features we computed for each trial the frequency power in different frequency bands (delta [1 4] Hz, theta [4 8] Hz, alpha [8 12] Hz, low beta [12 16] Hz). To compute these frequency power features, the EEG signals were first filtered in each band using a 250-order windowed sinc FIR-filter. Then, for each band, EEG signals were spatially filtered using two pairs of Common Spatial Patterns (CSP) filters [48]. Here, a regularized CSP with automatic covariance matrix shrinkage was used, as recommended in [49]. The resulting spectrally and spatially filtered signals were then squared, averaged over the epoch duration, and log-transformed [47], to obtain 16 frequency power features (4 CSP filters × 4 frequency bands). For the synchronized pipeline, the signal power was thus averaged in the time window from 0 to +0.6 s in order to match with the ERP based features, whereas for the non-synchronized pipeline, they were averaged over the whole duration of each 2-s long epochs.

For the two pipelines, to classify the extracted feature vectors, we used a shrinkage Linear Discriminant Analysis (sLDA) classifier, as recommended in [12,49]. We assessed the balanced classification accuracy for each subject, by using a stratified five-fold cross-validation procedure. For each step of the cross-validation, the CSP and sLDA were thus calibrated on four folds of the data, and tested on the remaining fifth one.

When combining ERP and frequency power features, we extracted each set of features independently, before concatenating all of them. Then, for each training fold of the cross-validation, we selected the top 20 features from them using minimum Redundancy Maximum Relevance (mRMR) feature selection [50], to keep an appropriate dimensionality for the classifier. These 20 features were then used as inputs of the sLDA classifier.

## 3. Results

### 3.1. ERPs

We found a load condition × type of sound × electrodes significant interaction (*p* < 0.01). This effect was due to higher P300 amplitude for the target sound in the low load compared to the high load condition on Pz, P4, and Oz electrodes (*p* < 0.001)—see Figure 4).

#### Frequency Analyses

We found a load condition × electrodes significant interaction with higher alpha [9–12] Hz power spectrum density over Pz electrode and higher low-theta [4–6] Hz power spectrum density over Oz electrode in the low load condition than in the high load condition (*p* < 0.01).

### 3.2. Single-Trial Classification Results

As regards the estimation of the mental workload using ERPs and event related frequency features, the mean accuracy was 50.4% (SD: 1.8) when using the ERP features only, 63.1% (SD: 9.5) when using the event related frequency features and was 50.4% (SD: 1.8) when combining the ERP and the event related frequency features (see Figure 5). As regards the estimation of the mental workload using the frequency features computed over the continuous signal, the mean classification accuracy reached 70.8% (SD: 12.5)—see Figure 5. Table 1 summarizes the classification results for the different pipelines and features.

In order to perform a neurophysiological interpretation of what the machine learning algorithms have learned from the data, we studied which frequency bands and channels proved the most informative. It should be noted that we only performed that analyze for the frequency features, as the other models gave chance level performances, and were thus uninformative. To do this analysis, we estimated the different features contributions by interpreting the sLDA and CSP weight vectors, using the method described in [51]. More precisely, for each subject and each fold of the cross-validation accuracy, we computed the sLDA activation pattern (forward model) from the trained sLDA weight vector. The feature with the largest absolute activation weight was the most informative feature. That feature corresponds to the power in a given frequency band, for a specific CSP spatial filter. We thus computed the activation pattern of this CSP filter (still using the method in [51]), and noted which EEG channel had the largest absolute activation weight. This thus gave us the most informative frequency band and channel for that subject and cross-validation fold. We repeated that procedure for all folds and subject, and counted the bands and channels that were the most often the most informative. Overall, by decreasing order of contribution, the most informative bands were 1–4 Hz, 4–7 Hz, 12–16 Hz and then 8–12 Hz. Regarding channels, still by decreasing order of contribution, the most informative ones were Pz, Oz, P3, P4, Fz, and then Cz.

## 4. Discussion

The motivation of this paper was to demonstrate the feasibility to assess pilot’s workload with a highly portable six dry-electrode EEG system in real flight condition. We designed a scenario in which pilots had to perform a low load and a high load traffic patterns along with a secondary oddball task. The first step of this study was to determine whether classical features (i.e., ERPs and PSD) could be extracted from the signal and allowed to discriminate the low vs. high load scenarios at the group level. The second step of this demonstration was to process off-line single trial classification to discriminate low vs. high load scenarios.

Firstly, the time-domain analysis at the group level disclosed that the P300 amplitude for the target sound was lower in the high load condition than in the low load condition on three electrodes out of six. These results are in line with [52,53] who demonstrated that the probe evoked P300 amplitude was negatively affected when flying task difficulty increased in simulated condition, leaving few available resources to process the auditory sounds. These findings may have interesting implication to better understand failure of pilot’s auditory attention under demanding settings such as inattentional deafness to auditory alarms that have been reported during critical flight phases [6,7,54,55]. These results thus confirmed that our dry-electrode EEG system actually measured brain signal as revealed by typical P300 response with higher differences on parietal sites [23]. More importantly, they show that passive auditory probes could be successfully used as an indirect index of mental workload in real flight conditions. The frequency domain analyses also echoed with the time-domain ones. The statistical findings indicated lower alpha band power on Pz in the high load scenario than in the low load scenario. These results are consistent with existing studies that have reported such decrease in the alpha band power on parietal sites when task demand increases [3,16,18]. Though, theta band power is generally expected to increase over the frontal electrodes [3,18], we only reported lower theta band power on Oz electrodes in the high load condition compared to the low load condition. However this result is consistent with [56,57] who found that lower theta band on occipital sites reflects increased level of engagement and vigilance. Taken together these findings confirm that sparse electrode setups can be used to record EEG signals and estimate mental workload, albeit with a lower spatial resolution than a high-density EEG cap [26,58,59].

Our second objective was to perform single trial classification over the electrophysiological signals to discriminate the two load conditions. A first attempt was to estimate mental workload using both ERPs and associated time-locked frequency features similarly to [22]’s approach. The classification accuracy did not exceed chance level (50.4%). This is most likely due to the poor contribution of the time domain features, which did not exceed chance level either when used alone, contrary to frequency power features (which reached for pipelines 1 and 2, respectively, around 63% and 70% of accuracy). Globally, the results suggested that ERP features are much more sensitive to noise—which is substantial in this real flight condition study—than frequency power features. This could be explained by the fact that ERP features represent the amplitude of a single time point in a single electrode. Thus a single artifact (e.g., a motion artifact) could severely affect several features at once and make them completely uninformative. Such noise and artifact could also cause large outliers in ERP features, which could in turn lead to defective classifiers, if trained on such features. Indeed, a strong artefact could turn a single or more ERP features into outliers, thus making the resulting feature vector an outlier itself (independently of the values of the other non-artefacted features in that feature vector). This outlier may thus bias the mean and covariance estimation of the corresponding class training data, thus resulting in a shifted LDA classifier. This shifted classifier may thus be unable to classify non-artefacted test data, as they would have a different mean and covariance. This seemed to be the case here, since using ERP features systematically led to chance level performances, independently of whether frequency features were used. In contrast, a single frequency power feature represents EEG signals combined across space (due to CSP spatial filtering) and time (due to averaging over the epoch). Thus, noises and artifacts that are localized in time and space would have their contributions substantially diminished by the averaging and spatial filtering operations. Moreover, the frequency power computation included a log-transformation, which thus reduced the contribution of large values, such as outliers. This may thus explain why frequency power features appeared as much more robust to noise and more effective in this realistic context than ERP features. Another explanation to this low classification performance could be that our protocol did not allow to collect enough trials to train a robust classifier especially in such a noisy environment. This is particularly true as we only used six electrodes, thus preventing us from using advanced signal processing technique such as ICA. Our protocol and results can not allow us to identify which is the most critical factor (motion artifacts, cockpit noise, reduced number of electrodes) affecting the outputs. However, one must conclude that dry-electrode technology still needs hardware improvements before it can be used during real flight operation. Indeed, safety is critical in aeronautics and lack of BCI reliability could trigger spurious assistance and thus impair global pilots’ performance. Moreover, the design of the dry-electrodes themselves have to be improved to offer better comfort for the pilots who will have to wear them over long period of time.

This work has several limitations. One first limitation of this study was that we did not counterbalance the order of the scenario. All the pilots started in the low load/pilot monitoring condition and then in the high load/pilot flying one. We acknowledge that this may have affected our results and could have induced states of higher arousal during the first traffic pattern and cognitive fatigue during the second one. Nonetheless, the ERPs and frequency statistical results confirmed that the first traffic pattern was associated with typical markers of lower arousal (higher alpha and theta band power) compared to the second one. Moreover, the two traffic patterns duration was too short (i.e., 1000 s duration) to induce fatigue or cognitive fatigue [36]. However, we can’t exclude learning effect and we agree that future experiments should counterbalance these two experimental conditions. A second limitation of this work is that we could not control for all the variables such as wind conditions as these experiments were conducted under realistic settings. We were also unable to conduct the experiment at the same time of day due to weather conditions, aircraft availability and safety, which severely constrained our planning. Despite these limits, our results at the group level were consistent with the neurophysiological literature. Moreover, single trial classification was performed at the individual level thus allowing to compare brain responses to the low and high load conditions in the same meteorological/aerological settings. A third limitation of this study is that we used a oddball paradigm as an indirect probe to assess mental workload. This approach was motivated to assess the feasibility to measure ERPs in a noisy cockpit environment and used them as features for workload classification. Such an approach is not viable for real-flight operations as it would require to trigger repetitive sound that could distract the pilots. The last limitation of this work is that one has to consider that the workload was not stationary in each leg of the flight pattern (namely: take off, crosswind, downwind, base and final) [60]. However, our goal was not to compare each of these legs especially as long as the duration of these legs are not equal, thus making difficult to perform statistical comparison across these legs without having the same number of data points. Moreover, the crosswind and base legs are quite short ( 30 s) meaning that very few auditory trials could be available for ERPs analyses. We believe that our approach is valid as long as the first and second traffic patterns include the same legs. Nonetheless, future work should include finer-grained analyses.

## 5. Conclusions

To the authors’ best knowledge, this study was the very first to report a pilot’s mental workload estimation with a six-dry-electrode EEG system in real flight conditions. The ERPs and frequency findings at the group level indicated that it is possible to quantify brain responses to variation of task demand. The single trial classification results were encouraging as long as frequency features were considered. Using such an approach allowed to reach more than 70% of accuracy to discriminate the two flying conditions defined as a sufficient accuracy for BCI [61]. These results open promising prospects to monitor the brain performance with very few electrodes in highly ecological settings. It confirms previous findings [34,38] that signal processing technique such as ASR and its Riemannian version [43] can help to improve the signal to noise ratio and classification accuracy, even in mobile recording scenarios. Notably, ASR and rASR are online processing methods which allow complex signal processing applications on low-cost hardware in everyday sitations outside of the lab. They are fast and enable an unsupervised procedure for passive BCI systems without the need of extensive training data. The next steps will be to perform on-the-fly mental workload estimation for a typical crew composed of a pilot flying, who is actually handling the trajectory and of a pilot monitoring who is in charge of supervising the flight parameters and communicating with the air traffic controllers [1]. Such an approach would allow to optimize task allocation based on each pilot’s workload. Eventually, one interesting perspective will be to use electrodes-around-the-ear technology [26] or ear-electrodes [62] that could fit into the pilot’s headset and offer optimal comfort for the pilots.

## Figures and Tables

**Figure 1 sensors-19-01324-f001:**
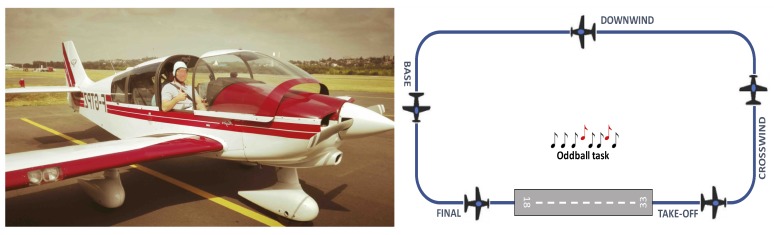
**Left**—ISAE-SUPAERO DR400 aircraft at Lasbordes airfield. **Right**—Experimental scenario: the pilots had to perform two traffic patterns (low and high load) along with an auditory oddball task.

**Figure 2 sensors-19-01324-f002:**
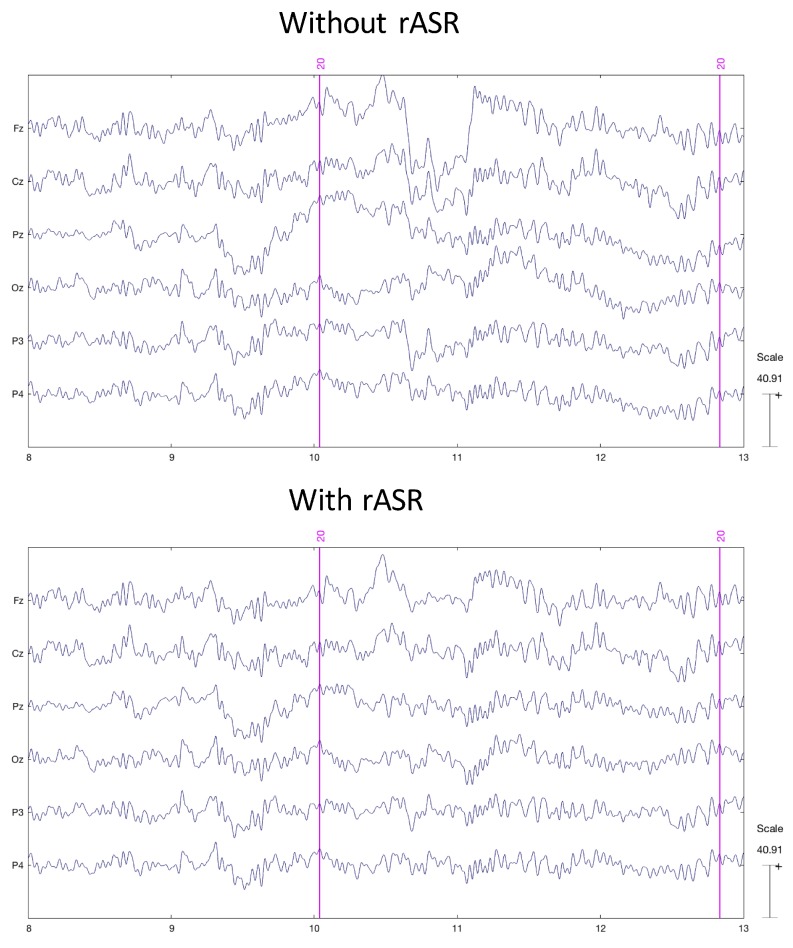
Up—Sample of EEG data before rASR processing for one subject. Sample of the same EEG data after rASR processing.

**Figure 3 sensors-19-01324-f003:**
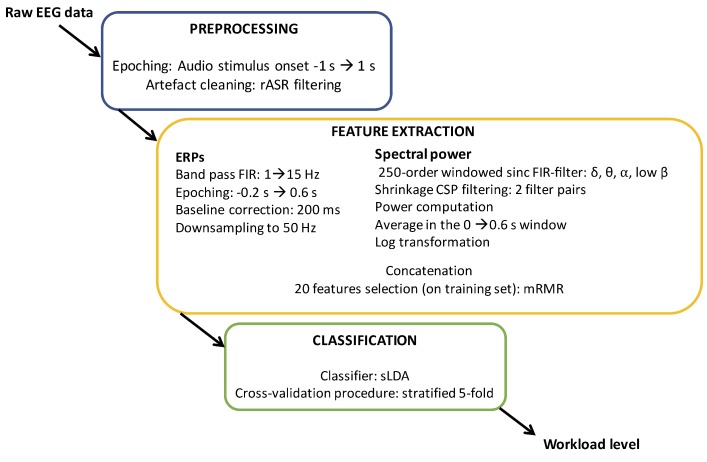
Illustration of the first processing pipeline with ERPs and frequency features. The second pipeline is identical to the first one to the exception that only frequency features were computed over successive and non overlapping epochs of two seconds.

**Figure 4 sensors-19-01324-f004:**
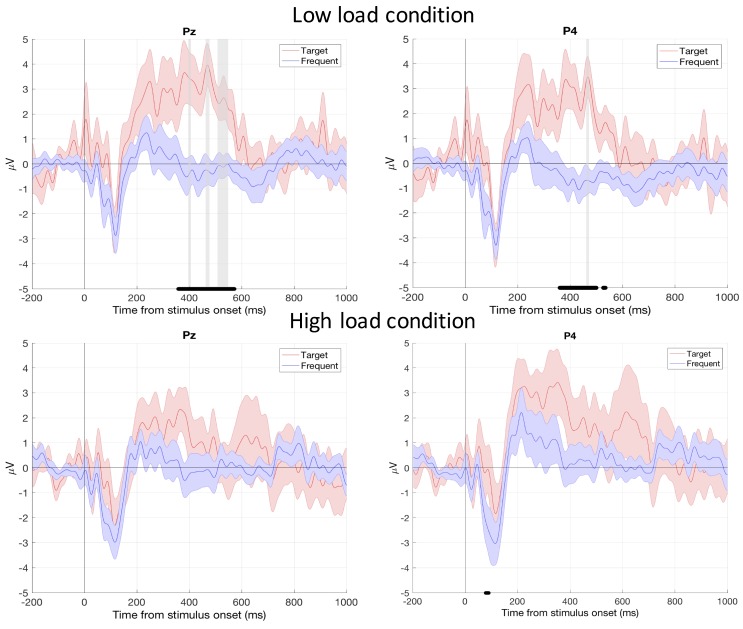
Grand averaged waveforms of the ERPs for parietal electrodes with standard error (shapes). The black lines on the *x* axis specify the time range when the target sound-related and the frequent sound-related ERP amplitudes were significantly different (*p* < 0.01). Up: low load condition. Down: high load condition. The vertical grey bars indicate when the P300 amplitude on the auditory target was statistically higher in the low load compared to the high load condition (*p* < 0.001). P300 considered time window was [350 600] ms.

**Figure 5 sensors-19-01324-f005:**
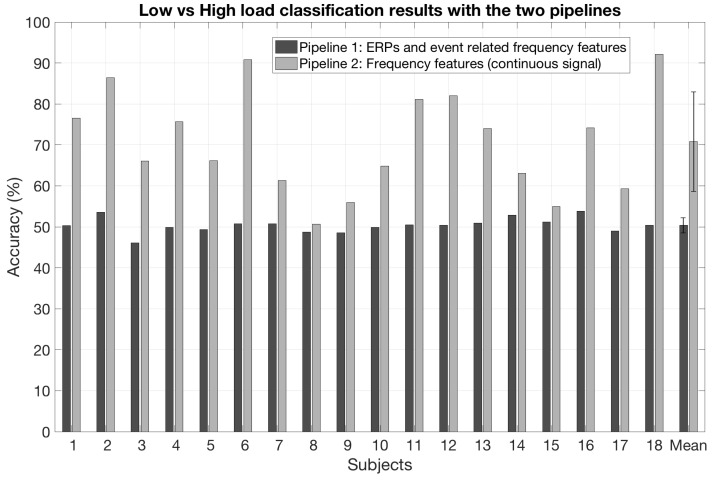
Single-trial classification results with the two pipelines for the 18 participants.

**Table 1 sensors-19-01324-t001:** Single trial classification results for the different pipelines.

Pipeline	Mean Accuracy	Mean Sensitivity	Mean Specificity
#1: ERPs & frequency	50.4%	51.2%	49.6%
#1: ERPs	50.4%	50.9%	49.9%
#1: frequency	63.1%	61.7%	64.5%
#2: frequency	70.8%	70.6%	71%

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
