# Peer review of "Monitoring Pilot’s Mental Workload Using ERPs and Spectral Power with a Six-Dry-Electrode EEG System in Real Flight Conditions"

_sensors, 2019, doi:10.3390/s19061324_

Round 1
Reviewer 1 Report
The authors present a study assessing or even better monitoring mental workload of pilots flying in real flight conditions. This is an interesting topic that has been studied before, however, the authors provide a more "easy to fit" solution using only six electrodes. My main concerns/ suggestions follow:
Main concerns
1) The authors conclude that hardware improvement is needed before it can be used for everyday flight operations. This is mainly due to noisy signal acquisition but the whole study could be more informative if there was any additional conclusion related to the limited number of electrodes or even the analysis procedure proposed. In essence, it is not clear what is the most critical factor affecting the output. Please comment on this and include it to the manuscript.
2) The authors make use of dry electrodes, known of limited signal quality, and at the same time they use only six electrodes limiting the applicability of ICA. To counterbalance this they use ASR in an attempt to remove short-time high-amplitude artifacts. However, such an approach may also affect the high frequency content of the signal, i.e., beta and gamma bands that are found to be associated with high cognitive demands. Hence, this possibly affects the inability of the study to identify significant differentiation in higher than theta bands.
3) The authors should include a figure depicting the actual and ASR-corrected EEG signal, since it is correctly stated in the conclusion that using online processing methods limits the possibilities and increase the difficulty.
Specific comments
4) Section 2.4.2: The authors use the wavelet transform to extract time-frequency information. Since, they applied rASR as a prepossessing step why not use just Short time fourier transform and have the frequency bands defined in the exact frequency limits and not pseudofrequencies (as translated from the wavelet transform).
5) Section 2.4.3:
a) For the single trial classification, since it is actually completed offline why not just select artifact free epochs instead of applying rASR just to be sure that there is no significant signal information loss?
b) In addition, since there are many steps involved in the single trial analysis, it would be nice to have a flowchart/ block diagram depicting the process, as well as adequate reasoning for each step involved.
c) What is the physical meaning (if any) of the final features selected? Could we draw a conclusion on the nature of features providing more information? Is it the frequency? Which band? What electrode/ brain region?
6) Section 3.1.2: Since the same individuals are performing both the "easy" and "difficult" task a 2 repeated measures ANOVA or include random factor effects would be the ideal choice.
Syntax
1) Line 205: "on three electrodes out six" --> "on three electrodes out of six"
Author Response
Dear Dr. Fanny Fang,
Ref: [Sensors] Manuscript ID: sensors-435298 -Monitoring pilot's mental workload using ERPs and spectral power with a 6 dry-electrode EEG system in real flight conditions
The authors are very grateful for the Reviewers’ suggestions and comments that have significantly helped to improve the paper’s scientific quality. In the following pages, we address the Reviewers’ comments in greater detail. Hopefully, this revision will meet your concerns.
Best regards
Frédéric Dehais
Reviewer 1
The authors conclude that hardware improvement is needed before it can be used for everyday flight operations. This is mainly due to noisy signal acquisition but the whole study could be more informative if there was any additional conclusion related to the limited number of electrodes or even the analysis procedure proposed. In essence, it is not clear what is the most critical factor affecting the output. Please comment on this and include it to the manuscript.
This reviewer raised an important point. We draw such a conclusion for two main reasons. The first reason is that certification standards are high in the aeronautical domain. One cannot implement a BCI that would not meet an accuracy superior to 99% as generally required in aeronautics. Indeed, a lack of BCI reliability could trigger spurious assistance and thus impair global pilots’ performance. In our case, a mean classification rate of 70 % cannot guarantee that the interaction would be adapted in an accurate manner. However, our protocol and results can’t allow us to conclude which of the factors had the most deleterious effect on the output. We can only acknowledge that it is due to a combination of the noise from the environment of the cockpit and motions artifacts, the use of dry electrodes that are known to have lower signal-to-noise ratio in comparison to gel electrodes and the limited number of electrodes that prevented us from using ICA. Eventually, the experiment duration was really short (less than 20 minutes) thus leading to collect a limited number of samples/epochs for the classification process. The second reason, is that the design of the dry electrodes need to be improved to offer better comfort if used over long period of time (e.g. 10 hours) as it is generally the case during real flight operations. These comments have been included in the penultimate paragraph of the discussion section.
The authors make use of dry electrodes, known of limited signal quality, and at the same time they use only six electrodes limiting the applicability of ICA. To counterbalance this, they use ASR in an attempt to remove short-time high-amplitude artifacts. However, such an approach may also affect the high frequency content of the signal, i.e., beta and gamma bands that are found to be associated with high cognitive demands. Hence, this possibly affects the inability of the study to identify significant differentiation in higher than theta bands.
ASR is a statistical method that learns properties of resting EEG during a calibration phase. Any data segment that contains strongly deviant signal properties is detected as being artifactual and therefore corrected. Physiological signals like oscillations or event-related potentials will not deviate from a resting EEG signal by an amount that is detected by ASR as artifactual. Several studies have shown that ASR can be used to correct data during motion while retaining oscillations as well as other cortical signals of interest: Arad, Bartsch, Kantelhardt, Plotnik (2018), Luu, Brantkey, Nakagomem Zhu, Contreras-Vidal (2017). We added this comment and the two last references in the penultimate paragraph of the introduction section.
That being said, we ran statistical analyses in the upper Beta [16 30] and gamma band [30 50]. The 2-way ANOVA disclosed a load condition x electrodes significant interaction with higher Beta Band [16-30] power spectrum density over Fz electrode in high load condition than in the low load condition. We also obtained higher single trial accuracy (75%) when including the beta and gamma power spectrum features in the classification pipeline. However, we decided not to include these results in the first version of our manuscript as one may say that upper beta and gamma bands also account for higher motor activity in the high load compared to the low load condition.
The authors should include a figure depicting the actual and ASR-corrected EEG signal, since it is correctly stated in the conclusion that using online processing methods limits the possibilities and increase the difficulty.
We thank the reviewer for this relevant comment. We added a figure (see Figure 2) that illustrates a sample of EEG data before/after rASR processing.
Section 2.4.2: The authors use the wavelet transform to extract time-frequency information. Since, they applied rASR as a prepossessing step why not use just Short time fourier transform and have the frequency bands defined in the exact frequency limits and not pseudofrequencies (as translated from the wavelet transform).
Indeed, we could have used Short Time Fourier Transform (STFT) as well. However wavelets, and in particular Morlet wavelets such as the ones used here, have been shown to be adapted to model and analyze EEG signals. They were even shown to be superior to STFT for classification for motor imagery BCI, see the paper below. This is why we used them in this work.
N. Brodu, F. Lotte, A. Lécuyer, "Comparative Study of Band-Power Extraction Techniques for Motor Imagery Classification", 2011 IEEE Symposium on Computational Intelligence, Cognitive Algorithms, Mind, and Brain (SSCI'2011 CCMB), pp. 1-6, 2011
For the single trial classification, since it is actually completed offline why not just select artifact free epochs instead of applying rASR just to be sure that there is no significant signal information loss?
Our motivation was to show that an “automatic” processing pipeline could be implemented to perform classification in an on-line manner. To that end, we decided to apply the same pipeline for each participant with the same settings and not to use a posteriori expert knowledge to select artifact free epochs by hand. Furthermore, many artifacts hampering the interpretation of neural signals are not detectable in the time domain.
In addition, since there are many steps involved in the single trial analysis, it would be nice to have a flowchart/ block diagram depicting the process, as well as adequate reasoning for each step involved.
This is an excellent suggestion and we have now integrated a flowchart to illustrate the processing pipeline (see figure 3).
What is the physical meaning (if any) of the final features selected? Could we draw a conclusion on the nature of features providing more information? Is it the frequency? Which band? What electrode/ brain region?
Indeed, interpreting what the machine learning algorithms have learned from the data could certainly bring some useful insights. Concerning the physical meaning of the features selected, please kindly note that feature selection method was only applied to the combined ERP+oscillations classification approach, due to the increased number of input features for that model. However, this model gave chance level performances, so it would not be relevant to interpret the selected features. Since the classification model based on oscillatory features gave by far the best classification performance, this means the frequency band power were the most information features. Then, it would certainly be interesting to see which frequency bands and channels proved the most informative. To do so, we estimated the different features contributions by interpreting the sLDA and CSP weight vectors, using the method described in (Haufe et al, 2014). More precisely, for each subject and each fold of the cross-validation accuracy, we computed the sLDA activation pattern (forward model) from the trained sLDA weight vector. The feature with the largest absolute activation weight was the most informative feature. That feature corresponds to the power in a given frequency band, for a specific CSP spatial filter. We thus computed the activation pattern of this CSP filter (still using Haufe et al approach), and noted which EEG channel had the largest absolute activation weight. This thus gave us the most informative frequency band and channel for that subject and cross-validation fold. We repeated that procedure for all folds and subject, and counted the bands and channels that were the most often the most informative. Overall, by decreasing order of contribution, the most informative bands were 1-4 Hz, 4-7Hz, 12-16Hz and then 8-12Hz. Regarding channels, still by decreasing order of contribution, the most informative ones were Pz, Oz, P3, P4, Fz, and then Cz. We added this new analysis to the revised manuscript.
Haufe, S., Meinecke, F., Görgen, K., Dähne, S., Haynes, J. D., Blankertz, B., & Bießmann, F. (2014). On the interpretation of weight vectors of linear models in multivariate neuroimaging. Neuroimage, 87, 96-110.
Section 3.1.2: Since the same individuals are performing both the "easy" and "difficult" task a 2 repeated measures ANOVA or include random factor effects would be the ideal choice.
We actually ran 3-way repeated measures ANOVA for the ERPs analyses (load x type of sound x electrodes) and a 2 way repeated measures ANOVA for the frequency analyses (load x electrodes).
Line 205: "on three electrodes out six" --> "on three electrodes out of six"
This has been corrected in the revised version of the manuscript.
Reviewer 2 Report
COMMENTS
Generally, this work represents potentially an interesting contribution to the literature. The main concern is that not enough information is provided on how this was implemented and also additional analysis could nicely complement this interesting work.
First, in sections 2.3 and 2.4, the mental workload of the subjects was likely not stationary in each leg of the flight pattern; this issue was not considered. Also further information about the quality of the data should be investigated. Namely, it would be informative to show the results in each leg of the flight pattern.
Also, the reliability of rASR in the recovery of noisy EEG sections should be demonstrated since here it is central to this analysis. Generally, the calibration part of EEG would be recorded on ground whereas the noisy contamination of EEG is obtained in flight leading to two very different conditions. Maybe this approach could be employed to check the quality of the recovered EEG.
In section 2.3, uncontrollable variables and controlled variables of the tasks should be explained. For example, any difference in testing environment for all subjects such as other aircraft traffic in the airfield, weather conditions (particularly wind and visibility), testing time (in the morning, afternoon, and evening), any behavioral instruction like no caffeine intake prior to 24 hours of the testing, flight instruction (e.g., airspeed, vertical speed, altitude, angle of bank) to the subjects, noise level in the cockpit, etc. should be provided.
Also, the trial duration in the secondary test should be provided.
A critical element is how the auditory stimuli was delivered to the subject in the cockpit? Did this possibility interfered with usual on-going communication? Would this be applicable to a real flight in presence of on-going communications? More information should be provided regarding this point.
Also, I was wondering if any subjective scale such as Cooper-Harper and NASA-TLX to show the validity of the level of challenge in the study was employed?
In section 2.4.1, the location of the ground and reference should be provided? The impedance levels of all the electrodes; at least the mean and standard deviation values should be provided. If wireless technology was employed please could you comment on any signal loss during recording? How the EEG and auditory stimulus were synchronized? What was the calibration protocol required for rASR, and what software version of rASR, EEGLAB, and MATLAB were employed?
In section 2.4.2, all the parameters of your two filters as well as a wavelet should be described. For example, FIR/IIR, filter order, attenuation level, window size, number of waves, etc. should be provided. In line 127 and line 133, it is indicated that noisy portions of data were removed however ASR removes high-variance principal components (the outputs of PCA) and reconstructs the signal with the remaining components, but do not remove EEG signal itself. Please could you the authors clarify?
In line 129, the burst criterion is set to 70, which is the standard deviation cutoff. This seems a very high value since generally ASR criterion are much smaller and often are not set beyond 6. Please could you the authors clarify this choice?
The calibration protocol for the ASR should be described as well as explain denoising technique for ERP.
In section 2.4.3, the description of the two processing pipelines is a bit unclear. Please provide more information about the denoising technique employed for the calibration part of EEG and if this part was recorded on ground or in flight. If this calibration part was not denoised, how the quality of the ASR results was assessed?
The numbers and figures comparing before and after ASR would be informative to add.
Please provide other parameters for the FIR filter beyond the low and high cut-off frequency.
What is the major difference between the first pipeline and the method that was used in section 2.4.2 except for the filtering parameters (0.5 to 30 Hz, vs. 1 to 15 Hz), temporal range (-0.2 to 1 s vs. 0 to 0.6 s), and downsampling to 50 Hz? Please could you clarify?
In line 153, it is mentioned: “the samples were used as features for classification.” Did the authors use all data points from 0 to 0.6 s by labeling each data point or did they extract any particular features from it, and used them with labels?
In line 160 indicates “averaged over the trial duration”, however it was mentioned that the second processing pipeline was executed regardless of stimuli onset in line 147. These two statements seems contradictory. Should the averaging be implemented over the epoch? Please could you clarify this point?
The use of the CSP for these data may be delicate due to the spatially unbalanced distribution of the electrodes – particularly due to Fz; the Cz, Pz, Oz, P3, and P4 will make a balanced diamond shape. Is there a possibility that CSP may produce biased results not only to Fz, but also to the other electrodes? Please could you comment on this?
The results would be stronger if some other classifiers would be considered beyond the LDA allowing to compare the results with these various classifiers.
It would be informative to indicate if the input attributes were Gaussian and each attribute had the same variance.
In addition, no mention of standardization or normalization for the feature sets before training were mentioned thus possibly resulting in biases by a set of attributes.
In section 3.1, what are the parameters for bootstrapping (e.g., number of bootstrap) and what was the difference with and without applying bootstrapping?
In section 3.1.1, please provide the values such as F-value, degrees of freedom. Were any main effect of load condition, type of sound, or electrodes obtained? Those should be reported even if not statistically significant. Also, in the method section the ANOVA should be described in details (level, test assumptions, etc.).
In Figure 2, the same scale for all the subgraphs should be used. Please indicate how many trials were included in these calculations and if any were dropped. Please indicate the statistical criterion for “The black bars specify … significantly different.” And also provide the time window for the P300.
In section 3.1.2, the use of a figure or table would be welcome.
In section 3.2, please could the authors show the confusion matrix and other quantifiable measures related?
In section 4, it would be very informative to show the rank or contribution of each features to the classification accuracy.
Also, the effects of a single artifact and/or outliers on the classification performance should be discussed.
Finally, although I understand that this work is more technical in nature, it would be good to briefly indicate what are the possible neural mechanisms underlying the changes in ERP and in spectral results.
I hope that this review was helpful to enhance this scientific report.
With best regards.
Author Response
Dear Dr. Fanny Fang,
Ref: [Sensors] Manuscript ID: sensors-435298 -Monitoring pilot's mental workload using ERPs and spectral power with a 6 dry-electrode EEG system in real flight conditions
The authors are very grateful for the Reviewers’ suggestions and comments that have significantly helped to improve the paper’s scientific quality. In the following pages, we address the Reviewers’ comments in greater detail. Hopefully, this revision will meet your concerns.
Best regards
Frédéric Dehais
Generally, this work represents potentially an interesting contribution to the literature. The main concern is that not enough information is provided on how this was implemented and also additional analysis could nicely complement this interesting work.
We sincerely thank the reviewer for his/her positive feedback about our study.
First, in sections 2.3 and 2.4, the mental workload of the subjects was likely not stationary in each leg of the flight pattern; this issue was not considered. Also, further information about the quality of the data should be investigated. Namely, it would be informative to show the results in each leg of the flight pattern.
We agree that a traffic pattern includes 5 different legs (namely: take off, crosswind, downwind, base and final) that are known to elicit different level of mental workload (scannella et al, 2018). However, our goal was not to compare each of these legs. One issue is that the durations of these legs are not equal making difficult to perform statistical comparison across these legs without having the same number of data points. Moreover, the crosswind and base legs are quite short (~30 seconds) meaning that very few auditory trials could be available for ERPs analyses. For these reasons, we decided not to use any triggers/markers to discriminate these legs. However, we believe that our approach is valid as long as the first and second traffic patterns include the same legs. Nonetheless, the reviewer made a relevant point and we now state in the conclusion section that future work should include finer-grained analyses (see last paragraph of the discussion section).
Also, the reliability of rASR in the recovery of noisy EEG sections should be demonstrated since here it is central to this analysis. Generally, the calibration part of EEG would be recorded on ground whereas the noisy contamination of EEG is obtained in flight leading to two very different conditions. Maybe this approach could be employed to check the quality of the recovered EEG.
The calibration data are indeed crucial for the successful correction of EEG data. Recording the calibration data in a resting scenario with the same subject is the proposed method to calibrate the ASR method and has been described and evaluated by other workgroups and the original authors of the ASR algorithm (Chang et al., 2018, Pion-Tonachini et al., 2018, Mullen et al., 2015). As stated in the manuscript (cf section 2.4.3), the first 30s of EEG signal were used to calibrate rASR. We agree that we did not clearly specify that this 30 s of cleaned signal was recorded on the ground before starting the experimental protocol while the participant was seated in the aircraft. We added this clarification in the revised version of the manuscript (see section 2.4.1).
In section 2.3, uncontrollable variables and controlled variables of the tasks should be explained. For example, any difference in testing environment for all subjects such as other aircraft traffic in the airfield, weather conditions (particularly wind and visibility), testing time (in the morning, afternoon, and evening), any behavioral instruction like no caffeine intake prior to 24 hours of the testing, flight instruction (e.g., airspeed, vertical speed, altitude, angle of bank) to the subjects, noise level in the cockpit, etc. should be provided.
The reviewer is right. We can’t control for all the variables when conducting experiments in real-life situation as wind conditions can slightly differ from one day to another. We were also not able to conduct the experiment at the same time of the day as weather conditions, availability of the plane and the flight instructor strongly constrained our planning. Despite these limits, our results at the group level were consistent with the neurophysiological literature. Moreover, single trial classification was performed at the individual level thus allowing to compare brain responses to the low and high load conditions in the same meteorological/aerological settings. This point is now discussed in the last paragraph of the discussion section. Pilots were instructed not to take any caffeine or medication prior to 24h of the testing. We added a sentence in section 2.1
Performing traffic patterns is a highly standardized task for which pilots are trained. We believe it is not relevant to detail these long procedures that might lengthen the text, confused the reader without providing key information to understand this study.
Also, the trial duration in the secondary test should be provided.
The duration of each traffic pattern (low load and high load) lasted 500 s.
A critical element is how the auditory stimuli was delivered to the subject in the cockpit? Did this possibility interfered with usual on-going communication? Would this be applicable to a real flight in presence of on-going communications? More information should be provided regarding this point.
The reviewer is right and this information was missing. The experimental computer was used to present the audio stimuli to the pilot’s aviation headset (Clarity AloftPro) through the auxiliary input. The sound intensity of the stimuli and background environmental noise in the cockpit were measured with a sound level meter. As for a previous experiment (Callan, et al, 2018), the sounds were presented at 85 dBA. The reported real-ear tested attenuation characteristics of the Clarity Aloft headset we used is reported to be 29 dB. The sounds of the stimuli were attenuated during radio communication but remained perfectly audible. The approximate signal to noise ratio was 35 dB with engine one. These clarifications have been added in section 2.3. One goal of our experiment was to assess whether we could pick ERPs from the brain signal in a noisy cockpit environment and potentially used them as features for workload classification. However, we believe that no one would like to use such an approach during everyday operations as it requires to trigger repetitive sound. We added details about the procedure to present audio stimuli in the material and method sections and discussed the limits of using ERPs for real flight operations in the last paragraph of the discussion section.
Also, I was wondering if any subjective scale such as Cooper-Harper and NASA-TLX to show the validity of the level of challenge in the study was employed?
We ran preliminary experiments with 4 pilots to design the experimental scenario and collected NASA-TLX score that confirmed that the 2 conditions obviously elicited two different levels of mental workload (high load condition=6.7, SD 0.45; low load condition 2.56, SD=0.75). We did not collect NASA-TLX score during the experiment. We added this precision in section 2.3
In section 2.4.1, the location of the ground and reference should be provided? The impedance levels of all the electrodes; at least the mean and standard deviation values should be provided. If wireless technology was employed please could you comment on any signal loss during recording? How the EEG and auditory stimulus were synchronized? What was the calibration protocol required for rASR, and what software version of rASR, EEGLAB, and MATLAB were employed?
As stated in the manuscript, we used the dry EEG Enobio system (Neuroelectrics). This system includes CMS and DRL electrodes that are used as reference electrodes. Dry EEG system does not provide impedance level but “offset level”. We carefully checked that the offset level for each subject was within the margins recommended by the manufacturer before starting the experiment as we could not take the risk to decrease the signal-to-noise ratio knowing that these experiments are costly. We used Lab Streaming Layer libraries (LSL, Swartz Center for Computational Neuroscience, UCSD, November 2018) to synchronize the oddball task in Matlab (Ver. 2017.b) with the Enobio acquisition software (NIC V2.0). All the EEG analyses were ran using the Matlab EEGLab toolbox (V14.1.2) on Matlab (2017.b). These precisions have been added in section 2.4.1. As stated in section 2.1, we only experienced data synchronization issues for four participants out of 22 – for the other participants, no data loss or synchronization occurred during the flights. The version of rASR and link to download the toolbox were already indicated in the first version of the manuscript. The explanation of the calibration procedure was detailed to the Reviewer (see the Reviewer 2’s second comment).
In section 2.4.2, all the parameters of your two filters as well as a wavelet should be described. For example, FIR/IIR, filter order, attenuation level, window size, number of waves, etc. should be provided. In line 127 and line 133, it is indicated that noisy portions of data were removed however ASR removes high-variance principal components (the outputs of PCA) and reconstructs the signal with the remaining components, but do not remove EEG signal itself. Please could you the authors clarify?
We used a windowed-sinc FIR filter with an order of 250. We have made the section on rASR more specific. The toolbox clean_rawdata in which the rASR algorithm is implemented contains several functions to correct artifacts, remove noise and faulty channels. In this section, two different functionalities of the toolbox were used and are now explained in more detail in section 2.4.2.
In line 129, the burst criterion is set to 70, which is the standard deviation cutoff. This seems a very high value since generally ASR criterion are much smaller and often are not set beyond 6. Please could you the authors clarify this choice?
This parameter has been shown to deviate strongly in different data sets: Chang et al., 2018. The original authors of the ASR algorithm evaluate in this paper the role of this parameter and reach a recommendation of a value between 30 and 70, depending on the data. The original recommendation of 5-10 turned out to be too aggressive for most use cases and removed/corrected more data than necessary.
The calibration protocol for the ASR should be described as well as explain denoising technique for ERP.
We believe that we already addressed this issue in our previous responses (see Reviewer’s 2nd question: calibration was done using 30 s of cleaned data recording on the ground while the pilote was seated in the plane).
In section 2.4.3, the description of the two processing pipelines is a bit unclear. Please provide more information about the denoising technique employed for the calibration part of EEG and if this part was recorded on ground or in flight. If this calibration part was not denoised, how the quality of the ASR results was assessed?
As previously explained (see Reviewer’s 2nd question), data were recorded on the ground while the pilot remained seated in the plane. We cannot make sure that the calibration data is optimal for the performance of the rASR toolbox (i.e. not recorded in a calm room with a Faraday cage). This evaluation would be very interesting and would add to the growing literature dealing with ASR methods. However, it is beyond the scope of our current investigation to evaluate the role of the calibration data here.
The numbers and figures comparing before and after ASR would be informative to add.
We added a figure to show an example of a sample of data before/after rASR processing (see Figure 2).
Please provide other parameters for the FIR filter beyond the low and high cut-off frequency.
We believe that we already addressed this question in one of the previous reviewer’s comment. In particular, we now specify in the manuscript that we used a windowed-sinc FIR filter with an order of 250.
What is the major difference between the first pipeline and the method that was used in section 2.4.2 except for the filtering parameters (0.5 to 30 Hz, vs. 1 to 15 Hz), temporal range (-0.2 to 1 s vs. 0 to 0.6 s), and downsampling to 50 Hz? Please could you clarify?
We assume you refer to the ERP analysis, if so, these are the only differences, the rest of the pipeline in section 2.4.3 was only the subsequent classification with the sLDA.
In line 153, it is mentioned: “the samples were used as features for classification.” Did the authors use all data points from 0 to 0.6 s by labeling each data point or did they extract any particular features from it, and used them with labels?
After preprocessing the EEG signals (band-pass filtering, baseline removal and downsampling) we simply concatenated the resulting EEG samples amplitude values from all channels into a single feature vector for each epoch, and used this feature vector as input to the sLDA classifier, as classically done for ERP classification in BCI (Lotte 2014). We thus labelled each feature vector with the corresponding workload level, to train the sLDA classifier. This sLDA was then applied to classify the unlabeled feature vectors from the testing set. This had now been specified in the manuscript.
Lotte, F. (2014). A tutorial on EEG signal-processing techniques for mental-state recognition in brain–computer interfaces. In Guide to Brain-Computer Music Interfacing (pp. 133-161). Springer, London.
In line 160 indicates “averaged over the trial duration”, however it was mentioned that the second processing pipeline was executed regardless of stimuli onset in line 147. These two statements seems contradictory. Should the averaging be implemented over the epoch? Please could you clarify this point?
Indeed, the power values were averaged over the epoch duration. Thank you for noticing this writing mistake, this has been corrected in the text.
The use of the CSP for these data may be delicate due to the spatially unbalanced distribution of the electrodes – particularly due to Fz; the Cz, Pz, Oz, P3, and P4 will make a balanced diamond shape. Is there a possibility that CSP may produce biased results not only to Fz, but also to the other electrodes? Please could you comment on this?
Please kindly note that the CSP is a purely data-driven spatial filtering algorithm, which only considers the channel signals but completely ignore their actual physical locations. Indeed, the CSP does not know where the channels are located and does not need to know given its objective function. The CSP will find linear combinations of the original channel signals, such that the resulting signals will have a power that is maximally different between conditions (here low vs high workload). It will thus find such weights to give to each channels whatever the actual physical locations of the sensors (see, e.g., Blankertz et al, 2008 below). To the best of our knowledge, there is thus no theoretical reason why the sensors unbalanced placement would lead to any bias toward a specific channel. Our interpretation of the features contributing the most (see our answer to another of your comments below) confirmed this, since Fz was neither the most nor the least contributing channel. However, if the reviewer has additional theoretical insights regarding this possible bias, that we might have missed, we would be happy to consider them.
Blankertz, B., Tomioka, R., Lemm, S., Kawanabe, M., & Muller, K. R. (2008). Optimizing spatial filters for robust EEG single-trial analysis. IEEE Signal processing magazine, 25(1), 41-56.
The results would be stronger if some other classifiers would be considered beyond the LDA allowing to compare the results with these various classifiers.
This is a relevant comment. We indeed ran different algorithm (SVM, with or without spatial filters,…) but none of them gave betters statistical results than the combination of CSP with sLDA. This is consistent with Lotte et al studies and reviews (Lotte et al 2018, Lotte 2015) who shown that these were the most appropriate technique when few training data are available. Our comparisons did not bring new knowledge. We thus decided not to present these results.
Lotte, F., Bougrain, L., Cichocki, A., Clerc, M., Congedo, M., Rakotomamonjy, A., & Yger, F. (2018). A review of classification algorithms for EEG-based brain–computer interfaces: a 10 year update. Journal of neural engineering, 15(3), 031005.
Lotte, F. (2015). Signal processing approaches to minimize or suppress calibration time in oscillatory activity-based brain–computer interfaces. Proceedings of the IEEE, 103(6), 871-890.
It would be informative to indicate if the input attributes were Gaussian and each attribute had the same variance.
The band-power features were specifically log-transformed, which makes them Gaussian-like, see, e.g., Ramoser et al. below. Regarding the variance, the different features did not have the same variance. Note however that this is not an issue at all when using an LDA classifier, as this classifier internally estimates the mean and variance of all features, and weight them accordingly to define the final decision function. Thus, contrary to classifiers such as SVM, the LDA is indeed known to be insensitive to feature scaling. Moreover, LDA is fairly robust to deviation from normality (see Duda et al, 2001).
Ramoser, H., Muller-Gerking, J., & Pfurtscheller, G. (2000). Optimal spatial filtering of single trial EEG during imagined hand movement. IEEE transactions on rehabilitation engineering, 8(4), 441-446.
Duda, Richard O, Peter E Hart, and David G Stork. 2001. Pattern Classification. New York: Wiley.
In addition, no mention of standardization or normalization for the feature sets before training were mentioned thus possibly resulting in biases by a set of attributes.
Indeed, no normalization was performed. However, as mentioned just above, contrary to other classifiers, normalization is not needed why using an LDA as classifier, as this classifier actually estimates the mean and variance of each feature and use this information to compute the weight to give to each feature in the final decision function. LDA is thus insensitive to normalization (Duda et al, 2001).
In section 3.1, what are the parameters for bootstrapping (e.g., number of bootstrap) and what was the difference with and without applying bootstrapping?
Brain responses to the different load conditions and type of sounds were determined by means of bootstrap statistics using 10,000 iterations. Bootstrap resampling has advantages over parametric statistical tests in that it does not assume normal distribution and homoscedasticity of the value of interest or the error terms (Efron & Tibshirani, 1994).
In section 3.1.1, please provide the values such as F-value, degrees of freedom. Were any main effect of load condition, type of sound, or electrodes obtained? Those should be reported even if not statistically significant. Also, in the method section the ANOVA should be described in details (level, test assumptions, etc.).
Degrees of freedom represents the number of parameters that need to be estimated for the distribution. The bootstrap distribution is usually not assumed to follow a t/F distribution thus no degree of freedom could be computed. We did not report main effect because there are not relevant with regard to our research question and we decided to focus only on the interactions.
In Figure 2, the same scale for all the subgraphs should be used. Please indicate how many trials were included in these calculations and if any were dropped. Please indicate the statistical criterion for “The black bars specify … significantly different.” And also provide the time window for the P300.
The reviewer is right and we modified the figure accordingly. A mean number of 17 trials were dropped by participant. We also provide time window for the P300 (see legend – figure 4). We added this precision in the beginning of 3.1.1 section.
In section 3.1.2, the use of a figure or table would be welcome.
We believe that adding a figure would be redundant with the text (only two significant results are detailed).
In section 3.2, please could the authors show the confusion matrix and other quantifiable measures related?
This is a relevant comment and we now provide the mean accuracy, specificity and sensitivity (see table 1).
In section 4, it would be very informative to show the rank or contribution of each features to the classification accuracy.
This is a relevant comment, also asked by reviewer 1, and we have thus added an analysis of the contribution of the different features. In particular, we added the following paragraph to the revised manuscript:
“In order to perform a neurophysiological interpretation of what the machine learning algorithms have learned from the data, we studied which frequency bands and channels proved the most informative. It should be noted that we only performed that analyze for the frequency features, as the other models gave chance level performances, and were thus uninformative. To do this analysis, we estimated the different features contributions by interpreting the sLDA and CSP weight vectors, using the method described in [51]. More precisely, for each subject and each fold of the cross-validation accuracy, we computed the sLDA activation pattern (forward model) from the trained sLDA weight vector. The feature with the largest absolute activation weight was the most informative feature. That feature corresponds to the power in a given frequency band, for a specific CSP spatial filter. We thus computed the activation pattern of this CSP filter (still using the method in [51]), and noted which EEG channel had the largest absolute activation weight. This thus gave us the most informative frequency band and channel for that subject and cross-validation fold. We repeated that procedure for all folds and subject, and counted the bands and channels that were the most often the most informative. Overall, by decreasing order of contribution, the most informative bands were 1-4 Hz, 4-7Hz, 12-16Hz and then 8-12Hz. Regarding channels, still by decreasing order of contribution, the most informative ones were Pz, Oz, P3, P4, Fz, and then Cz.”
Also, the effects of a single artifact and/or outliers on the classification performance should be discussed.
Please kindly note that we already discussed that point in the discussion section, at least to some extent. We nonetheless extended that discussion and added the following brief clarification in the same section:
“Indeed, a strong artefact could turn a single or more ERP features into outliers, thus making the resulting feature vector an outlier itself (independently of the values of the other non-artefacted features in that feature vector). This outlier may thus bias the mean and covariance estimation of the corresponding class training data, thus resulting in a shifted LDA classifier. This shifted classifier may thus be unable to classify non-artefacted test data, as they would have a different mean and covariance.”
Finally, although I understand that this work is more technical in nature, it would be good to briefly indicate what are the possible neural mechanisms underlying the changes in ERP and in spectral results.
The reviewer is right as the main contribution is at the methodological level. Neural mechanisms underlying changes in ERPs and spectral power have been widely detailed for decades (see Polich et al 2005, Borghini et al, 2014). The main motivation of this paper was to assess whether we could extract such canonical responses in the frequency and time domain so as to check that the dry EEG system could actually pick up brain signals.
Reviewer 3 Report
This study was about monitoring pilot mental workload using 6 EEG system in real flight condition.
The study is excellent in many aspects. The topic is quite interesting and the potential applications are also of interest.
I consider the study a good paper for this journal but I would like to add some comments that authors should review.
1) I don't think we can talk about cortical load using a 6 dry electrodes cap and this is a limitation of the study.
2) The fact that the low load was always first is a very important caveat that was mentioned by authors not only because the arousal aspect but the learning effect.
3) The results section lacks more clear tables comparing the results. I would suggest rewrite the results section. Accuracy measurement needs to be explained and supported by some sort of tables showing more than percentages.
The intro and discussion sections together with the methods makes this paper a good work.
Author Response
Dear Dr. Fanny Fang,
Ref: [Sensors] Manuscript ID: sensors-435298 -Monitoring pilot's mental workload using ERPs and spectral power with a 6 dry-electrode EEG system in real flight conditions
The authors are very grateful for the Reviewers’ suggestions and comments that have significantly helped to improve the paper’s scientific quality. In the following pages, we address the Reviewers’ comments in greater detail. Hopefully, this revision will meet your concerns.
Best regards
Frédéric Dehais
This study was about monitoring pilot mental workload using 6 EEG system in real flight condition. The study is excellent in many aspects. The topic is quite interesting and the potential applications are also of interest. The intro and discussion sections together with the methods makes this paper a good work. I consider the study a good paper for this journal but I would like to add some comments that authors should review.
We sincerely thank Reviewer 3 for his/her very positive comments on our manuscript.
I don't think we can talk about cortical load using a 6 dry electrodes cap and this is a limitation of the study.
We share the Reviewer’s point of view that reducing the number of electrodes affect the sensitivity to assess mental workload. However, we believe that our statistical results (ERPs and frequency analyses) are consistent with the literature with regards workload measurement. It has been shown in recent years that sparse electrode setups can be used to record EEG signals, albeit with a lower spatial resolution than a high-density EEG cap:
“Recording setups using the cEEGrid (Debener et al, 2015) have shown to reliably record traditional event-related-potentials since due to effects of volume conduction an EEG signal that is recorded at a particular scalp location may best be regarded as reflecting a mixture of several sources that may be located close, or further away from a particular recording channel. Whether a particular EEG channel is sensitive to a particular neural source depends (among other aspects) on the distance between the source and the electrode, the orientation of the source relative to the electrodes, and the relative distance of the recording electrodes to each other. While superficial strong sources of radial orientation may be picked up by nearby electrodes, tangential sources may contribute stronger to more distant than nearby electrodes (Väisänen et al., 2008).” From Bleichner and Debener, 2017. In this paper, the authors show that ERPs and oscillations are recorded with the grid, which is a well-studied example of a low-density EEG recording. Other examples of low-density EEG recording are sleep studies (e.g. Campbell, 2009) or epilepsy screenings (e.g. Curia et al., 2014) which rely on sparse electrode placement due to the need of a longterm recording which is not feasible with a high-density cap.
We added a sentence and some of these references at the end of the second paragraph in the discussion section.
The fact that the low load was always first is a very important caveat that was mentioned by authors not only because the arousal aspect but the learning effect.
We are totally in accordance with the Reviewer’s comment and we believe that this is a strong limitation that we now mention at the end of the discussion section.
The results section lacks more clear tables comparing the results. I would suggest rewrite the results section. Accuracy measurement needs to be explained and supported by some sort of tables showing more than percentages.
We agree with the Reviewer’s comment that is similar to the one raised by the first Reviewer. We now provide more details with regards to features/electrodes selection and mean specificity/sentivity.
Round 2
Reviewer 2 Report
This new version was definitely improved with respect to the previous one. The authors carefully answered the questions raised in the previous report. In my opinion, this new version is suitable for publication in Sensors.
I hope that my review was helpful to enhance the overall quality of this scientific report.